# Biomimetic Cooling: Functionalizing Biodegradable Chitosan Films with Saharan Silver Ant Microstructures

**DOI:** 10.3390/biomimetics9100630

**Published:** 2024-10-17

**Authors:** Markus Zimmerl, Richard W. van Nieuwenhoven, Karin Whitmore, Wilfried Vetter, Ille C. Gebeshuber

**Affiliations:** 1Institute of Applied Physics, TU Wien, 1040 Vienna, Austria; nieuwenhoven@iap.tuwien.ac.at (R.W.v.N.); gebeshuber@iap.tuwien.ac.at (I.C.G.); 2University Service Centre for Transmission Electron Microscopy, TU Wien, 1020 Vienna, Austria; karin.whitmore@tuwien.ac.at; 3Institute of Natural Sciences and Technology in the Art, Academy of Fine Arts Vienna, 1090 Vienna, Austria; w.vetter@akbild.ac.at

**Keywords:** biomimetics, passive daytime radiative cooling, climate resilience, functional structures

## Abstract

The increasing occurrence of hot summer days causes stress to both humans and animals, particularly in urban areas where temperatures can remain high, even at night. Living nature offers potential solutions that require minimal energy and material costs. For instance, the Saharan silver ant (*Cataglyphis bombycina*) can endure the desert heat by means of passive radiative cooling induced by its triangular hairs. The objective of this study is to transfer the passive radiative cooling properties of the micro- and nanostructured chitin hairs of the silver ant body to technically usable, biodegradable and bio-based materials. The potential large-scale transfer of radiative cooling properties, for example, onto building exteriors such as house facades, could decrease the need for conventional cooling and, therefore, lower the energy demand. Chitosan, a chemically altered form of chitin, has a range of medical uses but can also be processed into a paper-like film. The procedure consists of dissolving chitosan in diluted acetic acid and uniformly distributing it on a flat surface. A functional structure can then be imprinted onto this film while it is drying. This study reports the successful transfer of the microstructure-based structural colors of a compact disc (CD) onto the film. Similarly, a polyvinyl siloxane imprint of the silver ant body shall make it possible to transfer cooling functionality to technically relevant surfaces. FTIR spectroscopy measurements of the reflectance of flat and structured chitosan films allow for a qualitative assessment of the infrared emissivity. A minor decrease in reflectance in a relevant wavelength range gives an indication that it is feasible to increase the emissivity and, therefore, decrease the surface temperature purely through surface-induced functionalities.

## 1. Introduction

In the contemporary world, which is characterized by ever-increasing resource consumption and an escalating waste crisis, the biologization of technology offers revolutionary perspectives for materials science [1]. The concept of passive radiative cooling (PRC), inspired by natural processes, exemplifies how leveraging biogenic materials can provide sustainable and innovative solutions for technical applications. van Nieuwenhoven et al. [2] highlight Nature’s ability to produce materials that, while not always exhibiting extreme properties, prioritize longevity, repairability, biodegradability and reusability. These attributes are invaluable today, facilitating a sustainable approach to resource management. With these ecological principles in mind, this work aims to provide the groundwork for a low-cost and simple solution to an increasingly relevant problem: The cooling of indoor environments.

### 1.1. Increased Indoor Environment Cooling Demand

Recent reports by the International Energy Agency (IEA) state that indoor environment cooling (IEC) is currently responsible for 9% of the total electricity consumption [3]. Due to climate change and an overall increase in living standards, related cooling energy consumption is estimated to more than double by 2050 without intervention. On the other hand, only 15% of the 3.5 billion people who live in hot climates had access to IEC in 2021, with even lower access levels in Sub-Saharan Africa and South Asia. The lack of access to IEC puts much of the global population at an increased risk for heat stress, adversely affecting thermal comfort, labour productivity and human health [4]. Moreover, traditional IEC solutions, such as air conditioning (AC), cause problematic side effects. As long as fossil fuels contribute significantly to power generation (65% in 2016, worldwide average [5]), the operation of AC further accelerates climate change. AC also directly heats up its immediate environment. Heat exchangers transport heat from an indoor environment to the outside environment. Studies estimate that anthropogenic heat from AC can increase the temperature in urban areas by 0.5 °C to 1.5 °C [6,7,8]. Passive radiative cooling (PRC), more precisely passive daytimeradiative cooling (PDRC), could address these challenges with energy-free operation and the riddance of excess heat directly into outer space. A cheap, simple and high-impact realization of PDRC would offer a way for lower-income regions of the world to address the increasing impact of climate change.

### 1.2. Passive Daytime Radiative Cooling

Passive daytime radiative cooling (PDRC) as a research field has gained increasing traction and promises considerable benefits to traditional IEC solutions. Sub-ambient temperatures of a PDRC surface exposed to the sun require two decisive properties of the surface: high reflectivity of the incident solar radiation and high emissivity in the infrared region. Figure 1exemplifies this. To clarify, the emissivity of the surface of a material is its effectiveness in emitting energy as thermal radiation. It ranges from 0 (no emitted radiation) to 1 (perfect emitter). The wavelength of the incoming solar radiation that reaches the Earth surface ranges from around 290 nm to 3 µm. The reference solar spectrum AM 1.5 (commonly used to characterize the performance of solar cells under standardized conditions) specifies a power density of 1000 Wm^−2^ [9].

Outgoing radiation (i.e., infrared blackbody radiation) can only escape to cold space through a wavelength region called the atmospheric window, which ranges from 8 µm to 13 µm. Outside of the atmospheric window, most of the mid- and far-infrared radiation is absorbed by the atmosphere. The atmospheric window limits the emitted power to 100–150 Wm^−2^ for a surface temperature of 300 K [10].

To achieve effective PDRC, a surface should not only achieve a solar reflectance much higher than 0.9 but should also exhibit a high emissivity value (>0.9) in the atmospheric window [11]. High emissivity should be confined to wavelengths within the atmospheric window. Emissivity and equivalently absorption [12] outside this range allow radiation from thermal sources, such as ambient air or nearby structures, to be absorbed by the PDRC surface, thereby diminishing its cooling efficiency [13]. If narrow-band emissivity is not feasible, the PDRC surface should only face upwards to reduce incoming radiation from nearby infrared emitters. Extensive mathematical analyses of PDRC can be found in the literature [10,11,13,14,15,16].

**Figure 1 biomimetics-09-00630-f001:**
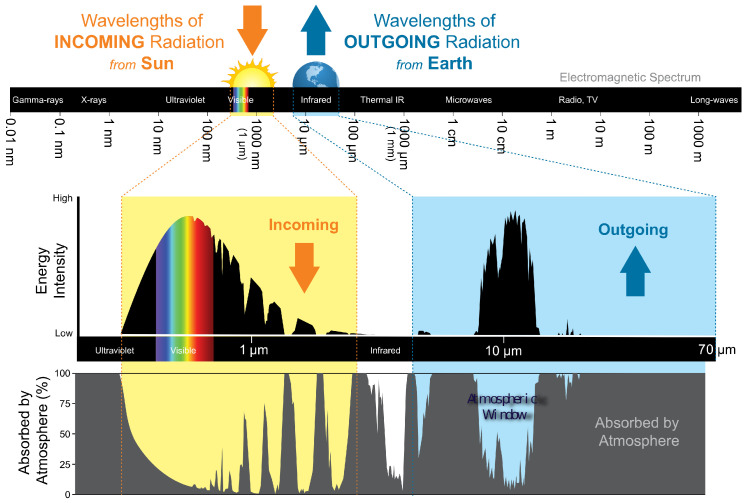
Incoming and outgoing radiation energy intensity and the absorption spectrum of the atmosphere. The bulk of the outgoing energy lies within the atmospheric window from 8 to 13 µm [17].

### 1.3. State-of-the-Art PDRC Solutions

Several approaches exist to realize PDRC, such as a multilayered HfO_2_ and SiO_2_ photonic cooler [18], a metamaterial cooler created by embedding resonant polar dielectric microspheres randomly in a polymeric matrix [19], a monolayer of silica, a SiO_x_N_y_ layer and a metal reflective layer [20] and a porous polymer coating [21]. Liang et al. [16] gave a comprehensive review of recent developments and applications of PRC. While these technical solutions utilize multilayered structures, randomly distributed particle structures or porous polymer photonic structures, numerous organisms have independently evolved mechanisms for self-cooling through PDRC. The literature describes a variety of butterflies [22,23,24], ants [12,14] and beetles [25,26]. These insects cool themselves with various refined functional structures on the surface of their bodies or wings. There have been multiple attempts to transfer PRC properties from organisms to engineering [12,27,28]. However, to our best knowledge, none of these attempts concentrated on transferring the functionality to biological materials, such as chitin, chitosan and keratin. The functional structures in butterflies that induce PDRC are complex due to their multi-functionality and would be hard to replicate with conventional methods [24]. Saharan silver ants, with their relatively simple triangular hairs, were considered more promising for the development of replication methods.

### 1.4. Saharan Silver Ants

The Saharan silver ant (*Cataglyphis bombycina*) is a genuinely fascinating insect that evolved various methods to deal with the harsh climate in the desert. These ants are one of the few animals that can withstand the midday sun and, consequently, avoid predators such as lizards, who avoid going out during the day. The silver ant body produces heat shock proteins before leaving the nest [29]. The silver ant exhibits remarkable navigational skills and is among the fastest known animal species relative to its body size [30,31]. The exoskeleton surface of the silver ant, which features triangular-shaped chitin hairs, is particularly interesting (Figure 2a). These hairs provide PDRC to the ants, by contributing to the reflective silver color of these ants and enhancing their infrared emissivity in the wavelength region of the atmospheric window [14]. Small corrugations on the top sides of the triangular hairs enhance the reflection of solar light due to Mie scattering. The flat undersides of the triangular structures and a low refractive index layer beneath them (air gap) result in the total reflection of the solar radiation (Figure 2b) [14]. The emitted infrared radiation has a larger wavelength than the triangular structures. This causes the triangular shape to provide a gradual change in the refractive index (from chitin to air). A gradual change in the refractive index facilitates high absorption and equivalently, high emissivity [12]. Notably, the PDRC of the triangular hairs functions effectively across a wide range of incident angles. This characteristic is essential for successful technical implementation (i.e., transferring the structure of the triangular hairs to a material that is used to cover the surface of bodies that are to be passively radiatively cooled, such as house facades), ensuring that the surface functionality remains independent of the angular position of the sun.

### 1.5. Utilizing Chitin and Chitosan as PDRC Materials

Freigassner et al. [32] explored the potential of the biomaterials chitin and keratin. They are abundant biopolymers found in nature and are discussed for sustainable applications due to their unique nanostructural properties and biodegradability. Chitin, found in arthropod exoskeletons and fungi, provides mechanical stability through strong hydrogen bonds. High mechanical stability is essential for outdoor applications but causes challenges in transferring micro- and nano-structures to the surface of chitin. Chitin from various animal waste products can be transformed into sustainable, high-value products for different applications, highlighting the potential for efficient resource-recovery methods [33]. Chitosan is a water-soluble, chemically altered, deacetylated product of chitin that can be easily processed into a film for structural functionalization. This study concentrates on potential applications of chitin and chitosan for PDRC applications. Lauster et al. [34] highlighted the potential of chitosan and chitin as widely available biocompatible polymers for PDRC applications.

### 1.6. Low-Cost and High-Impact PDRC Solutions

This project aims to provide the groundwork for low-cost and high-impact PDRC solutions. These solutions are based on biomimetically structured bio-based materials, such as chitin. These materials are made from abundantly available chemical elements, are biodegradable, and can be synthesized under ambient conditions. This study focuses on chitin and chitosan due to their favorable properties concerning PDRC as well as biodegradability and local availability. The weathering experiments on shrimp shells provided insights into the weather resistance of chitin (Section 2.2). Chitosan offers greater flexibility for structural modifications compared to chitin. Therefore, the basic idea is to functionally structure chitosan and then chemically convert it to chitin to improve weather resistance. For a circular economy, chitin can be sourced from waste, chemically converted to chitosan for structural modifications, and then reverted to chitin. In the current manuscript, we describe the initial steps taken to reach these goals: To test the feasibility of the structural modification of chitosan films, the color-inducing microstructural gratings of a compact disc (CD) are transferred onto such films (Section 2.4). This demonstrates the possibility of introducing functionality through structures on chitosan films. Subsequent experiments focused on the transfer of the silver ant’s surface structure onto chitosan films (Section 2.4). The quality of the transfer was evaluated with microscopy and FTIR spectroscopy (Section 2.5).

## 2. Materials and Methods

This study investigates the structural modification of chitosan films through the transfer of microstructures, assessing the outcomes via microscopy and FTIR spectroscopy, while also examining the effects of weathering on chitin.

### 2.1. Chitin

Shrimp shell waste was collected from a local market (Naschmarkt Stand 177-178, 1060 Vienna, Austria), rinsed, dried, and flattened under constant pressure for 72 h, yielding flat samples (Figure 3a). The pressure was exerted utilizing a water-filled glass container with a total mass of 1.7 kg. The shell samples were scratched with a sharp crystal mounted on a micromanipulator (Newport M-461 Series, XYZ micromanipulator). Fifteen parallel scratched lines with a separation distance gradually increasing from 10 µm to 100 µm, and a length of 1 cm were scratched onto the shrimp shell samples. The scratch depth was set to 10 µm, though variations may occur due to the irregular thickness of the shrimp shells.

### 2.2. Weathering

To assess the weatherability of chitin structures and, therefore, their suitability for outdoor applications (e.g., on house facades), the scratched shrimp shell samples were exposed to heating and cooling cycles in a climate chamber (ESPEC LHU-114 constant climate cabinet).

### 2.3. Chitosan

Chitosan is derived from chitin through deacetylation, resulting in a polymer that exhibits increased solubility. Chitosan is available in a highly purified form (e.g., from Sigma-Aldrich, St. Louis, MO, USA [35]) and as an inexpensive dietary supplement. Typically, the dietary supplement has a lower concentration. This study utilized the supplement product Greenfood Chitosan [36], which has a specified concentration of 90% chitosan. Chitosan films were produced using a straightforward and reproducible procedure conducted within a laminar flow fume hood to ensure a contaminant-free and safe operation environment. The powder contained in the dietary supplement chitosan capsules was dissolved in a dilute acetic acid solution (approximately 10% by volume: 5 g water, 500 mg acetic acid, 400 mg chitosan) under constant stirring at approximately 50 °C. This results in a homogeneous but very viscous solution. The viscous chitosan solution was then uniformly spread over a surface using doctor blading, which effectively yielded an even, thin film. The thickness of the film was set to 0.5 mm by employing raised bars on both sides of the sample area. These bars leave a slit of consistent thickness between the doctor blading and the surface. The coated surface was left to dry at room temperature, allowing the solvent to evaporate and the film to solidify. This production protocol produced chitosan films with a consistent thickness and uniform consistency.

### 2.4. Functional Structure Imprinting onto Chitosan

Two types of structures were transferred from the respective original to chitosan: Microstructured gratings from a CD were directly transferred to chitosan to demonstrate that chitosan can successfully accept the imprint of microstructures. By contrast, the transfer of silver ant microstructures required a polyvinyl siloxane (PVS) stamp as an intermediary because of its favorable release properties and the ability to preserve nano-sized structures [37]. PVS (Coltene President The Original Light Body, Coltène/Whaledent AG, Altstätten, Switzerland) is commonly used in dentistry to create precise impressions. For the direct transfer of the CD microstructured gratings, the protective layer of the CD was removed with adhesive tape. The exposed microstructured grating (responsible for a CD’s characteristic color reflections) is used as a platform onto which the dissolved chitosan is deposited and evenly distributed. The film is stripped off the CD after drying for 24 h.

The process of transferring the silver ant nanostructures was indirect: the round gaster (hind part) of the silver ant was embedded in a cardboard perforation to provide a reasonably flat surface for the PVS stamp production. A PVS drop of about 1 cm diameter was applied to this construction and allowed to dry for 10 min (Figure 4). The central area of the stamp (1 mm diameter) was thereby impressed with the negative of the silver ant’s body surface. The chitosan solution was evenly doctor-bladed onto the PVS stamp. After drying for 24 h, the chitosan film was stripped off the stamp with tweezers.

### 2.5. Imaging

#### 2.5.1. Scanning Electron Microscopy

The SEM model ThermoFisher Scios II was used to compare weathered and unweathered shrimp shells and examine a specific section of the silver ant specimen. A focused ion beam (FIB) integrated into the SEM system was employed to precisely obtain a cross-section micrograph of the silver ant’s triangular hairs for detailed analysis.

#### 2.5.2. Confocal Microscopy

A confocal microscope (Nanofocus µsurf explorer) was used to investigate the shrimp shell chitin samples before and after the climate chamber treatment, as well as the PVS stamp and chitosan films imprinted with the PVS stamp.

### 2.6. Spectroscopy

The Bruker Lumos FTIR microscope and spectrometer was used to investigate the chitosan films and compare the reflectance of structurally modified and unmodified samples. The measured reflectance spectrum ranges from wave number 7000 cm^−1^ to 400 cm^−1^ (which is equivalent to wavelengths from 1.4 µm to 25 µm). The visible spectrum is, therefore, not captured. The reflectance is given in relation to a reference background (standard gold mirror provided by the FTIR). For each sample, 20 spectra on different parts of the surface (split between modified and unmodified areas) with a pixel size of 200 µm × 200 µm were captured. The average spectrum and its standard deviation were calculated using the software Panorama 4.0 by LabCognition. The reflectance was measured because it allows us to draw conclusions about the emissivity of the surface if the specific criteria listed below are met. Kirchhoff’s law (Equation (1)) is applicable when the transmission is negligible. A sufficiently thick chitosan film exhibits negligible transmission, so Kirchhoff’s law applies in this context. Kirchhoff’s law states that emissivity and absorptance must be identical to avoid violating energy conservation. It is important to note that in this context, reflectance, absorptance and emissivity should more precisely be referred to as total reflectance/absorptance/emissivity. For example, total reflectance is the reflectance in all directions (into a hemisphere) for uniform incident radiance (from all directions, i.e., from a hemisphere) [38].
(1)ϵ=α=1−ρ
where ϵ is the total emissivity, α is the total absorptance and ρ is the total reflectance. The second part of Equation (1) is deductible from the conservation of energy (α+ρ=1) when transmittance is zero. Nicodemus [38] further states that this relation holds true not only for total quantities but also for directional quantities. Directional reflectance is the reflectance in all directions (into a hemisphere) for a collimated incident beam (from one direction). This statement is also reversible, i.e., directional reflectance is the reflectance in one direction for uniform incident radiance (from all directions). Measuring directional reflectance would allow the calculation of directional emissivity but requires the implementation of an integration sphere [39]. The Bruker Lumos used in this work utilizes a Cassegrain objective [40]. This means that incident radiation, as well as the reflected light, does not cover the whole hemisphere but only a ring section between 17°and 37° (with respect to the perpendicular direction). The specific values were provided by Bruker Support inquiry. We have to assume that the relation between directional reflectance and directional emissivity (e.g., a lower reflectivity means higher emissivity) holds for the reflectance as we measured it. Therefore, the conclusions drawn from the FTIR measurements are inherently preliminary, serving as an initial proof of concept to guide further investigation.

## 3. Results and Discussion

The experiments explored the resilience of bio-based materials, such as chitin shells and chitosan films, under environmental stress and their potential for functional applications. Scratched shrimp shells were subjected to weathering simulations, while chitosan films were imprinted with micro and nano-structures for optical and cooling purposes. Reflectance measurements were conducted to evaluate the performance of structured chitosan, compared to unstructured chitosan, with a focus on the suitability for passive cooling in architectural and technological contexts. Challenges related to the transfer of microstructures were addressed by optimizing fabrication techniques and providing insights into the practical implementation of bio-inspired materials for sustainable solutions.

### 3.1. Weathering Tests

If bio-based and biodegradable materials are to be used as components for house facades, they need to be weather-resistant. The climate chamber experiment was conducted to simulate the environmental conditions affecting the chitin material over time. To simulate a hot summer day, the temperature was gradually oscillated between +30 °C and +50 °C at relative humidity levels between 20% and 55%. One cycle lasted for eight hours and was repeated 30 times. For the simulation of a cold climate, the same cycle structure was used, but with temperatures ranging from −5 °C to +5 °C at a relative humidity between 40% and 80% (Figure 3b).

The appearance and dimensions of the scratches on the shrimp shells remained predominantly unchanged following exposure to the weathering cycles (Figure 5a).

### 3.2. Chitosan Imprinted with Reflective Structures

The chitosan film imprinted with CD microstructures exhibited iridescent colors, which demonstrates that the transfer of the microstructures from the CD to the chitosan film was successful. The reflection of sunlight in the chitosan film was captured with a digital camera (Canon EOS R10) (Figure 5b).

### 3.3. Chitosan Imprinted with Cooling Structures

Various challenges were encountered in the attempts to transfer the silver ant body microstructures via the PVS stamp to the drying chitosan film: chitosan and PVS repel each other, preventing close contact, which is necessary for the transfer of the structures. One of the attempts to address this challenge was to work with higher-viscosity chitosan films. Another attempt utilized the stamp as a passive part, so that the chitosan film was doctor-bladed onto the stamp rather than the stamp stamped onto the film. The successful attempt was the combination of both. The two most promising chitosan films were selected for subsequent FTIR measurements. The imprinted chitosan films featured the same general structure as the stamp but were less distinctive and had some speckles, which were not part of the stamp structure (Figure 6).

### 3.4. FTIR

The FTIR spectra of the two samples of chitosan films that were structured with silver ant micro- and nanostructures via the same PVS stamp exhibit slightly ambiguous results, within the margin of error, concerning the averaged reflectance for IR wavelengths, in the range measured (1.4 µm to 25 µm, see Figure 7). Both samples exhibited lower reflectance in the range of the atmospheric window. As discussed in Section 2.6, lower reflectance generally indicates higher emissivity, which our application desires.

## 4. Conclusions and Outlook

Replicating the cooling mechanism of the silver ants’ microstructures is a complex challenge. This study demonstrated a protocol for transferring CD microstructures and silver ant nano- and microstructures onto chitosan films. The FTIR setup used in this study was sensitive to various uncontrolled variables, including variations in specular and diffuse reflection, the macroscopic curvature of the examined surfaces (caused by the naturally curved silver ant body) and discrepancies between directional reflectance and the measured reflectance (see Section 2.6).

The preliminary FTIR results warrant further investigation to obtain robust and reproducible data. Enhancing the FTIR measurements using an integrating sphere [39] could improve accuracy by allowing for the measurement of hemispherical reflectance. Additionally, creating a flat, technical replica of the silver ant gaster with identical microstructures on a planar substrate could minimize the impact of macroscopic curvature on the measurements. This artificial silver ant gaster could be fabricated using nanolithography and, depending on the imprinting capability, could serve as either a negative stamp or a positive replica, necessitating a mediator like PVS.

The transfer process cannot replicate the underside of the triangular microstructures. Incorporating a total-reflection layer on the underside of the triangular structures is necessary to enable cooling over a wide range of incidence angles (Figure 2b). To achieve a total-reflection layer, an additional layer with a refractive index lower than that of chitin is necessary beneath the triangular microstructures. Chitin has a refractive index of $n_{\mathrm{chitin}}=1.61$ [41]. Possible candidates for this layer include air-filled gaps supported by structural frameworks or materials with lower refractive indices, such as calcium fluoride (n=1.43).

Weathering experiments in a climate chamber (ESPEC LHU-114) indicated a good durability of microstructures on chitin surfaces. However, comprehensive weathering tests need to be conducted to ensure compliance with ISO/DIN standards (e.g., ISO 2810, ASTM G7, AATCC TM186) [42]. Considering chitin’s superior environmental resistance compared to chitosan, future investigations will focus on transitioning micro- and nanostructured chitosan to chitin [34], with an emphasis on preserving the vital structural features required for effective passive daytime radiative cooling (PDRC).

We propose utilizing a rolling mechanism, analogous to a paint roller, to facilitate the imprinting of cooling micro- and nanostructures over larger surface areas. This roller would transfer the intricate silver ant cooling structures instead of pigments, enabling the large-scale production of functional films. The fabrication of larger structured cooling surfaces (several cm^2^) would enable the substitution of FTIR measurements with more reliable calorimetric assessments for evaluating cooling performance.

## Figures and Tables

**Figure 2 biomimetics-09-00630-f002:**
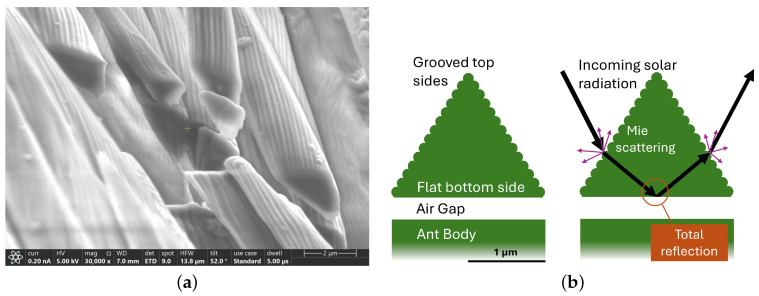
(**a**) SEM micrograph of cross-sections cut with a focused ion beam (FIB) through triangular chitin hairs of a Sahara silver ant gaster (hind part). Scale bar—2 µm. (**b**) Illustration of the triangular cross-section of a silver ant hair. Incoming solar radiation undergoes Mie scattering at the small indentations of the top sides. The light that enters the silver ant hair can be reflected on the bottom side when the conditions for total reflection are met (incidence angle and difference in refractive index between silver ant hair and air gap) [14]. Scale bar—1 µm.

**Figure 3 biomimetics-09-00630-f003:**
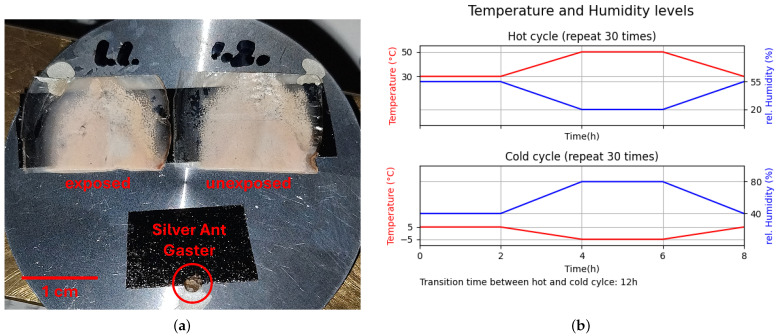
(**a**) SEM sample holder with exposed and unexposed shrimp shell sample, as well as silver ant gaster (rear segment of the silver ant). Scale bar—1 cm. (**b**) Climate chamber cycles (programmed).

**Figure 4 biomimetics-09-00630-f004:**
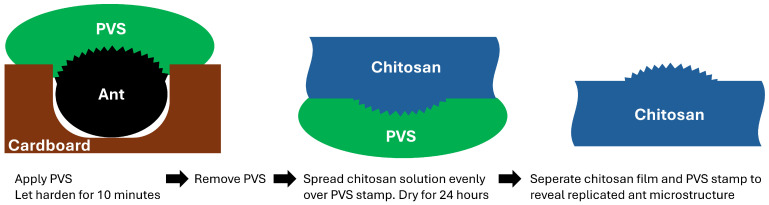
Process of creating a copy of the silver ant surface structure in chitosan with the help of a PVS stamp.

**Figure 5 biomimetics-09-00630-f005:**
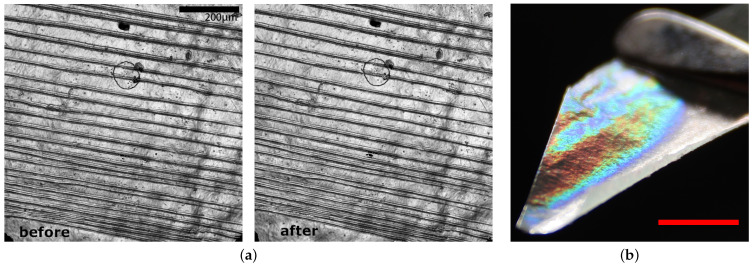
(**a**) Confocal image of scratched shrimp shell before and after exposure in the climate chamber. Scale bar—200 µm. (**b**) Chitosan film with iridescent microstructures transferred from a CD. Scale bar—0.5 cm.

**Figure 6 biomimetics-09-00630-f006:**
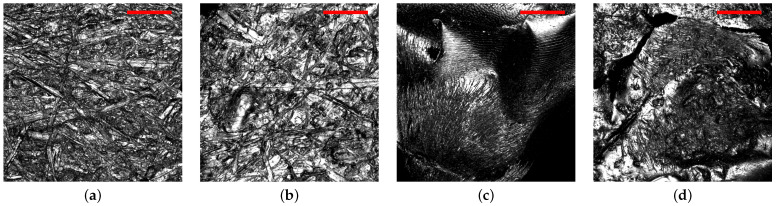
Confocal images of (**a**) an ’unstructured’ area of the PVS stamp, which shows the structure of the cardboard that surrounded the silver ant gaster. (**b**) The PVS–cardboard structure transferred onto chitosan. (**c**) A structured area of the PVS stamp structured with an ant gaster. (**d**) The PVS–ant structure transferred onto chitosan. Scale bars—200 µm.

**Figure 7 biomimetics-09-00630-f007:**
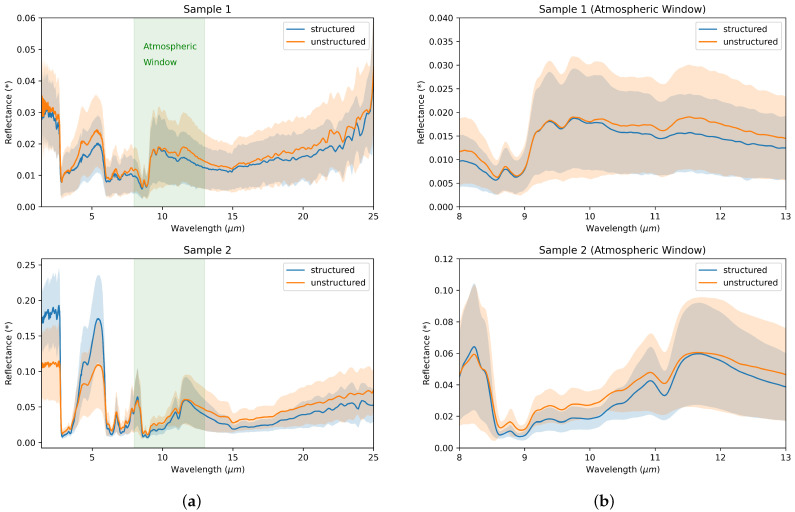
(**a**) Average reflectance (* in relation to a reference gold mirror) of structured and unstructured chitosan films. The structured areas in both samples exhibit a slightly higher reflectance for wavelengths greater than 6 µm. However, this difference is less than the calculated standard deviation. Below 6 µm, the two samples feature inconsistent differences in reflectance. (**b**) Zoom into the respective region of the atmospheric window in (**a**).

## Data Availability

All Data can be accessed at the Research Data Repository of TU Wien at https://doi.org/10.48436/tchcj-fam86, accessed on 15 October 2024.

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
