# Peer review of "Biomimetic Cooling: Functionalizing Biodegradable Chitosan Films with Saharan Silver Ant Microstructures"

_biomimetics, 2024, doi:10.3390/biomimetics9100630_

Round 1
Reviewer 1 Report
Comments and Suggestions for Authors
Dear authors!
The draft is devoted to the reconstruction of the microstructure of Saharan Silver Ant on the surface of chitosan. Quite an interesting article from an academic point of view. The manuscript has a relevant direction on the creation of coatings with passive cooling on the surface of polymer coatings. Unfortunately, the article is difficult to read and has a number of significant shortcomings, and until they are eliminated, it cannot be recommended for publication.
1) The introduction is written interestingly, but it takes up four full typeset pages (that's a lot). In places it contains completely irrelevant information, for example, point 1.2. Chitin vs. Chitosan – Why so much about chitosan? At the same time, the introduction does not provide an overview of the work on this topic: what is the structure of the ant's cover and why does it work so well? Are there similar coatings for passive cooling? What can your result be compared with?
2) There is no clearly formulated purpose of the work and scientific novelty.
3) It is unclear why it is put on a CD.
4) Why do the authors first cite studies with materials bought on the market and then jump to others - commercial chitosan.
5) Why did you use chitosan, it is not the best option? You provide the results of climate tests, but what will happen to the film (especially with microtexture) in a humid atmosphere? (The film will swell, change its geometric dimensions and it will be impossible to talk about any microtexture.)
6) How did you dry the chitosan films and their characteristics (thickness, moisture content)? You work with the acid form, but have you tried using the reduced form of the films?
7) The figures in the supplementary materials duplicate the figures in the article (of course, there are new figures). Authors should use supplementary materials when discussing the results.
8) In the end I did not understand whether the authors managed to achieve the result or not.
Reviewer 2 Report
Comments and Suggestions for Authors
The publication is missing a common thread. The different chapters are not really consistent and for the reader it is hard to understand the revelance. It remains unclear why a part of the story is focussed on scratched chitosan and why the topography of a CD was copied into chitosan. Regardig the measurements of reflection of IR-radiation at the relevant wavelengths it is not clear if the differences between smooth and structured surfaces are significant and what that would mean to practical application. Regardig practical application there is not clear if and how the ant's surface topography could be applied to technical products. As a conclusion I could only state, that the matter of increasing the solar reflection of surfaces is definitely an important one, and achieving hat by implementing a microtopographie is very interesting, but regarding this biomimetic approach a bit more systematic research (including measurements of angle- and wavelenght dependent reflection parameters) and a more coherent presentation of approach and results would be appropriate.
Reviewer 3 Report
Comments and Suggestions for Authors
The topic of the manuscript is important and interesting. It is recommended to check the following before publishing.
1. Introduce chitin biomaterials for the first time.
2. How can the incoming infrared radiation from other thermal emitters (such as air, nearby house facades, or high mountains) be minimized without decreasing the infrared emissivity in the atmospheric window region?
3. What was the purpose of the climate chamber experiment conducted on the chitosan material?
4. What challenges were encountered in the attempts to transfer the ant body microstructures to the drying chitosan film, and what were the solutions tried to address these challenges?
5. What was the range of infrared wavelengths measured in the FTIR analysis of the chitosan films structured with Saharan Silver Ant micro- and nanostructures? More explanation is needed.
6. What was the key finding from the FTIR analysis regarding the reflectance and emissivity of the structured chitosan films in the atmospheric window? More explanation is needed.
Round 2
Reviewer 1 Report
Comments and Suggestions for Authors
Dear authors!
I have carefully read the new version of the article and the authors' response to my comments: the draft of the article is well revised, in this presentation the purpose and ideas of the work become clear. This article is not a finished research work, but only the beginning of a series of follow-up work. It can be said that the authors have ‘bookmarked’ this direction of research in the field of creating coatings with passive cooling on the surface of polymer coatings (with imitation of biological structures). I believe that in this revision the article can be published.
I also suggest the authors to consider protecting the structured surface of chitosan using grafted transparent polymer coatings with hydrophobic properties, this can significantly increase the lifetime of chitosan films and materials.